# FINE-TUNING LARGE LANGUAGE MODELS FOR TEXT RANKING WITH LISTWISE CONSTRAINTS

## ABSTRACT

With the rapid adoption of large language models (LLMs) across diverse applications, retrieval augmentation has become a key factor for improving downstream performance. Recent advances show that LLM-based retrieval can substantially enhance ranking quality. In this work, we present a novel LLM-based retrieval framework optimized along three complementary dimensions: (1) a customized attention-based fusion of hidden-layer representations, (2) a dedicated multi-layer perceptron (MLP) module for enriched feature transformation, and (3) a new listwise learning objective, ListRank loss, to capture fine-grained relevance order. Experimental results demonstrate that our model achieves state-of-the-art performance. The model is publicly available for download on HuggingFace.

## 1 INTRODUCTION

Information Retrieval (IR) is the fundamental task of retrieving relevant documents given a textual query. With the rapid advancement of large language models (LLMs), IR has become increasingly crucial across a wide range of domains. In particular, modern large-scale question answering systems typically rely on retrieval-augmented generation (RAG)Lewis et al. (2020), where external knowledge bases or web documents are searched to provide relevant context for response generation. Beyond question answering, IR also plays a vital role in domains such as entertainment, finance, and academia, and can be regarded as a gateway technology for knowledge acquisition in modern computing.

The history of IR can be traced back to the 1950s, when Gerard Salton introduced the vector space model (VSM)Salton et al. (1975), representing documents with term frequency and combining it with Boolean retrievalSalton et al. (1983). This was followed by classical IR models that represented both queries and documents as vectors, with relevance measured by cosine similarity. In 1994, the BM25Robertson et al. (1995) algorithm was proposed, introducing a more refined term weighting scheme that remains a widely adopted baseline today. In the early 2000s, the rise of web search engines revolutionized retrieval, with Google's PageRankBrin & Page (1998) combined with textual matching dramatically improving search quality. After 2010, the advent of deep learningKrizhevsky et al. (2012) enabled the use of neural models for text representation and retrieval. The introduction of the TransformerVaswani et al. (2017) architecture in 2017 triggered a wave of high-performance encoders, such as the BERTDevlin et al. (2019) family, which soon became central in retrieval tasksKarpukhin et al. (2020).

The breakthrough of GPTRadford et al. (2019) models in 2022 sparked a global wave of LLM-based intelligent agents. Researchers found that scaling model size and training on large, high-quality datasets could yield dramatic improvements in reasoning and generalization. Consequently, retrieval research has shifted from encoder-based paradigms toward LLM-based approaches. For example, methods such as RankGPTSun et al. (2023), LRLMa et al. (2023b), and PRPQin et al. (2023) reformulate retrieval as a ranking task, leveraging the generative capabilities of LLMs to produce ranked outputs. Meanwhile, the RankLLaMA team (2023) introduced RepLLaMA and RankLLaMAMa et al. (2023a): RepLLaMA serves as an embedding model, where queries or documents are appended with a special token, and the hidden state of the token is extracted as the text embedding. The model is trained with InfoNCE lossOord et al. (2018) by maximizing the similarity of positive query-document pairs and minimizing that of negatives. RankLLaMA, on the other hand, functions as a re-ranking model, concatenating query and document, appending a special token, and

mapping the hidden state through a fully connected layer to obtain relevance scores, again trained with InfoNCE loss.

Beyond specific instantiations such as RankLLaMA, contemporary LLM-based reranking pipelines share several structural limitations. First, representation is commonly bottlenecked by single-token summarization (e.g., using a final or special token), which presumes that long-range semantics can be compressed into one vector. Second, the relevance head is often shallow (linear or single-layer), providing limited inductive bias to convert generative features into discriminative signals. Third, despite the intrinsically listwise nature of ranking, many training objectives remain pointwise or pairwise, leaving the global permutation structure underutilized.

We address these limitations with three synergistic components. We replace single-token summarization with attention-based fusion over all token embeddings to capture broader context. We adopt a stronger MLP head to better align features with discriminative scoring. We further introduce a listwise objective that encodes full-order constraints within each candidate set. Together, these yield ListRank, which consistently improves text retrieval and reranking performance.

## 2 METHOD

### 2.1 PRELIMINARIES

We consider the *reranking* task in a standard two-stage retrieval pipeline. Given a user query $Q$ and a candidate set of documents $C = \{D_1, D_2, \ldots, D_k\}$ produced by a coarse retriever, the reranker assigns each candidate $D_i$ a real-valued *relevance score* and sorts the candidates in descending order of these scores to produce the final ranking.

Formally, the reranker is a function

$$f_\theta : (Q, D_i) \mapsto s_i \in \mathbb{R}, \quad i = 1, \ldots, k,$$

where $(Q, D_i)$ is the **input** pair consisting of a query and a document, and $s_i$ is the **output** scalar score (a float) indicating the predicted relevance of $D_i$ to $Q$. Higher $s_i$ implies stronger relevance. Collecting scores over the candidate list yields $\mathbf{s} = [s_1, s_2, \ldots, s_k]$, which induces a permutation by sorting in descending order.

In implementation, the input pair $(Q, D_i)$ is encoded as a single sequence by concatenating the query and document with task-specific prompts or separators. The model computes a score $s_i$ for each pair independently (pointwise formulation) or jointly over the list (listwise formulation). In either case, the **inputs** are the query $Q$ and a candidate document $D_i$, and the **output** is a single real-valued score $s_i \in \mathbb{R}$ used for ranking.

Concretely, during training we concatenate the query and a document into a single sequence as follows:

$$\text{input} = \text{`query: } \{Q\} \text{ document: } \{D\}\text{'}$$

where $\{Q\}$ and $\{D\}$ denote the raw query and document texts inserted into a fixed template. The reranker consumes this sequence and outputs a scalar score $s_i$ for $(Q, D_i)$. For comparison, RankLLaMAMa et al. (2023a) similarly concatenates the query and document, and appends a closing token "" at the end of the sequence. An overview of the proposed model pipeline is shown in Figure 1.

For clarity, we distinguish the outputs in two phases. During inference, the model produces scalar scores $s_i$ for each $(Q, D_i)$, which are used to sort candidates. During training, the intermediate scores $\{s_i\}$ are transformed into per-position losses $\{L_i\}$ and aggregated by the ListRank objective into the final training loss.

### 2.2 BASE MODEL

We adopt **Qwen3** as the backbone for our reranking model. The Qwen3 familyYang et al. (2025) is a decoder-only Transformer language model trained with the standard autoregressive objective.

Concretely, given an input sequence constructed from a query–document pair (Section 2.1), Qwen3 LLM produces token-level hidden states

$$\mathbf{H} = [h_1, h_2, \ldots, h_n] = \text{Qwen3-LLM(input)},$$

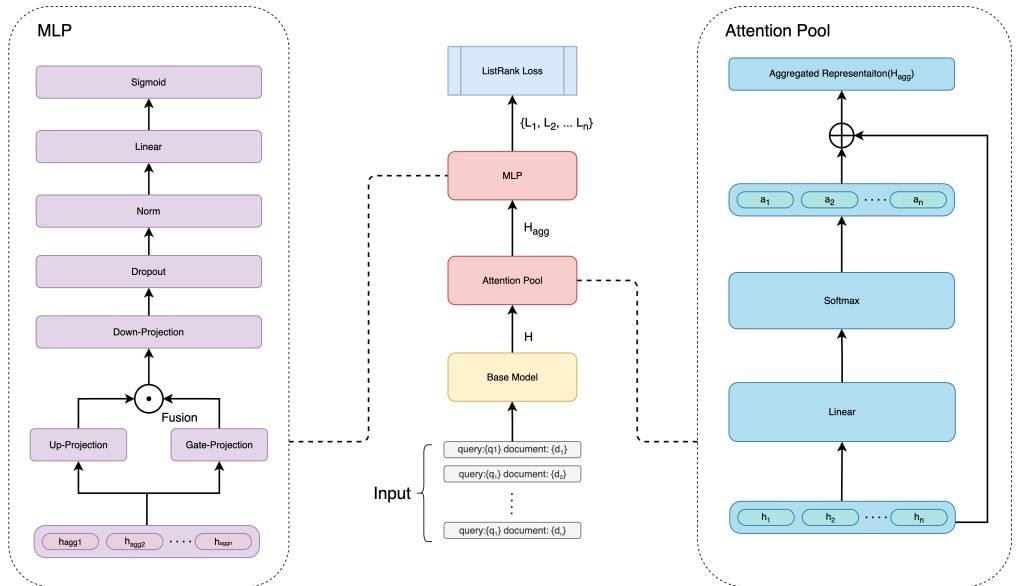

Figure 1: Overall pipeline of ListRank. A coarse retriever first returns top-$k$ candidates. For each $(Q, D_i)$, we construct a single input sequence "query: {Q} document: {D}" and feed it into the LLM backbone. Token-level hidden states are fused by the Attention Pool to form $\mathbf{H}_{\text{agg}}$, which is then transformed by an MLP into an intermediate score $s_i$. During training, these scores are converted into per-position losses $L_1, L_2, \ldots, L_n$ and aggregated by the ListRank objective as the final training output (loss), enforcing listwise ordering over candidate sets.

where each $h_i \in \mathbb{R}^d$ denotes the last-layer representation of the $i$-th token and $d$ is the hidden dimension. These representations are subsequently aggregated by the **Attention Pool** module (Section 2.2) to obtain an input-level embedding $\mathbf{H}_{\text{agg}}$, which is then transformed by the **MLP** head (Section 2.3) into an intermediate score $s_i$ for the pair $(Q, D_i)$.

## 2.3 ATTENTION POOL

In many reranking systems (e.g., BERT-based reranking), the LLM feature for a query–document pair is taken from the hidden state of a special token (e.g., [CLS]), implicitly assuming that the model can compress global context into a single vectorNogueira et al. (2019); similarly, RankL-LaMAMa et al. (2023a) concatenates the query and document and uses the sequence-final token ("" in LLaMA-style tokenization) to summarize the context. However, when processing long texts, attention sparsity and the reduced utilization of middle-context information make global context integration challenging, which can exacerbate hallucination risksMaynez et al. (2020). With the rapid growth of modern textual resources, documents are becoming significantly longer, making it critical to address the challenge of mitigating hallucination in semantic compression.

To alleviate the degradation of representation quality caused by long-text compression in LLMs, this work proposes a full-token feature fusion technique, which is primarily applied to the output of the reranking model, specifically at the similarity projection layer mentioned earlier. Concretely, the hidden representations of all tokens $\mathbf{H} = [h_1, h_2, \ldots, h_n]$ are first mapped into a token_num $\times 1$ vector through a fully connected layer. This vector is then normalized via a softmax function to obtain what we refer to as attention weights $\mathbf{A} = [a_1, a_2, \ldots, a_n]$. Each token's hidden representation is weighted by the corresponding attention weight, and the weighted features are summed to form an aggregated attention-based representation $\mathbf{H}_{\text{agg}}$. The computation process, which we refer to as the **attention pool**, can be formally expressed as follows:

$$\mathbf{A} = \text{Softmax}\Big(\text{Linear}(\text{Decoder}(\mathbf{input}))\Big)$$

$$\mathbf{H}_{\mathrm{agg}} = \sum_{i=1}^{n} a_i \cdot h_i = \mathbf{A} \cdot \mathbf{H}$$

### 2.4 Multi-Layer Perceptron

The **multilayer perceptron (MLP)**Devlin et al. (2019) has undergone extensive development and optimization since its introduction, demonstrating remarkable success across various domains. The advantages of MLPs can be summarized as follows:

1. **Nonlinear mapping** – through stacked linear transformations combined with nonlinear activation functions (e.g., ReLUNair & Hinton (2010), GELUHendrycks & Gimpel (2016)), MLPs project input features into higher-dimensional, more discriminative spaces.

2. **Feature fusion and transformation** – when the input consists of multi-dimensional feature vectors (e.g., token embeddings or sentence embeddings), MLPs can integrate these signals into more compact and information-rich representations.

3. **Regression or classification** – MLPs serve as effective output layers, mapping feature vectors to scalar scores (for regression) or probability distributions (for classification).

4. **Lightweight and flexible** – compared to large-scale Transformers with billions of parameters (e.g., 8B or 13B models), MLPs are computationally inexpensive, yet provide significant representational power.

Consequently, introducing a well-designed MLP module for similarity projection adds negligible computational cost relative to the backbone LLM, but yields substantial improvements in reranking performance by enabling a more accurate transformation of LLM features.

In this work, we adopt an MLP design inspired by the Qwen architectureYang et al. (2025). Specifically, the MLP employs a gated structure rather than a plain feed-forward network. The process begins with two parallel linear projections: (i) an up-projection that expands the feature dimension, and (ii) a gate-projection that produces a gating vector. The up-projection output is passed through a nonlinear activation function, such as SiLUElfwing et al. (2018) or HardSwishHoward et al. (2019), and is then element-wise multiplied with the gate-projection output to enable gated information control. The fused features are subsequently projected back to the hidden dimension via a down-projection, followed by dropout and normalization for regularization and training stability. Finally, a linear projection and sigmoid activation are applied to produce the scalar similarity score.

Formally, the computation can be expressed as:

$$h_{\mathrm{up}} = \phi(W_{\mathrm{up}}x), \quad h_{\mathrm{gate}} = W_{\mathrm{gate}}x$$

$$h_{\mathrm{fusion}} = h_{\mathrm{up}} \odot h_{\mathrm{gate}}$$

$$h_{\mathrm{down}} = W_{\mathrm{down}}h_{\mathrm{fusion}}$$

$$h_{\mathrm{out}} = \sigma(W_{\mathrm{final}} \mathrm{Norm}(\mathrm{Dropout}(h_{\mathrm{down}})))$$

where $\phi(\cdot)$ denotes the SiLU/HardSwish activation, $\odot$ is element-wise multiplication, and $\sigma(\cdot)$ is the Sigmoid function.

### 2.5 ListRank Loss

From a first-principles perspective, the reranking task requires *ordering* a set of top-$k$ candidate documents $C = \{D_1, D_2, \ldots, D_n\}$ to produce an accurate relevance ranking. RankLLaMA adopts the InfoNCE loss for pairwise discrimination, but this does not explicitly exploit the *listwise* ordering signals essential for fine-grained ranking.

To address this limitation, we propose the **ListRank loss**, which directly models listwise order information:

1. **Model output.** Given candidates $C$, the model produces scores

$$\mathbf{h}^{\text{out}} = \left\{ h_1^{\text{out}}, h_2^{\text{out}}, \ldots, h_n^{\text{out}} \right\}.$$

2. **Sub-list construction.** These scores are sequentially partitioned into $n - 1$ descending sub-lists:

$$\{h_1^{\text{out}}, \ldots, h_n^{\text{out}}\}, \ \{h_2^{\text{out}}, \ldots, h_n^{\text{out}}\}, \ \ldots, \ \{h_{n-1}^{\text{out}}, h_n^{\text{out}}\}.$$

3. **Cosine-based weights.** For the $i$-th sub-list (with total length $m$), an initial cosine weight is computed as

$$\cos_{\text{origin},i} = \cos\left( \frac{\pi}{2} \cdot \frac{i}{m} \right),$$

followed by softmax normalization:

$$\cos_{\text{weight},i} = \frac{\exp\!\left(\cos_{\text{origin},i}\right)}{\sum_{j=1}^{m} \exp\!\left(\cos_{\text{origin},j}\right)}.$$

4. **Sub-list loss.** For each sub-list, a temperature-scaled log-softmax loss is calculated:

$$\ell_i = -\log \frac{\exp\!\left(h_i^{\text{out}}/\tau\right)}{\sum_{j=1}^{m} \exp\!\left(h_j^{\text{out}}/\tau\right)},$$

where $\tau$ is a temperature constant.

5. **Loss sorting.** The sub-list losses $\ell_i$ are sorted in descending order to obtain $\ell_i^{\text{sort}}$.

6. **Final ListRank loss.** The overall objective is

$$\mathcal{L}_{\text{ListRank}} = \sum_{i=1}^{m} \cos_{\text{weight},i} \ \ell_i^{\text{sort}}.$$

The proposed ListRank loss directly models listwise ordering for reranking tasks, addressing the limitations of pairwise objectives like InfoNCE. By decomposing model scores into descending sub-lists and applying a temperature-scaled log-softmax, the loss captures both global and local ranking structures. Cosine-based weights emphasize top-ranked candidates, while sorting sub-list losses prioritizes the most challenging positions. This combination ensures precise top-$k$ ranking, stable training, and improved differentiation among closely scored documents, making ListRank particularly effective for high-precision retrieval scenarios.

Based on the Attention Pool, MLP optimization, and ListRank loss proposed in this work, we construct a new model, which we refer to as ListRank. The input to ListRank consists of a query concatenated with a list of documents ordered by relevance. Given this query–document input, the model produces relevance scores for each document. During training, the model is supervised using the ListRank loss to learn the listwise relationships present in the input data. At inference, the model scores the top-$k$ candidates selected by a coarse embedding-based retrieval stage, producing a fine-grained reranked list.

## 3 EXPERIMENTS

### 3.1 DATASET

We construct our training data from the MS MARCO passage ranking corpus. To accommodate the listwise objective proposed in this work, we employ a RankGPT-refined subset of the MS MARCO passage ranking dataset, where candidate passages are carefully re-ranked to provide high-quality listwise supervision. To ensure experimental fairness, the models trained with the ListRank loss mentioned below are trained on a combination of the constructed MS MARCO list dataset and the pairwise MS MARCO original dataset, whereas the models trained with InfoNCE are trained only on the pairwise MS MARCO original dataset. This is because InfoNCE lacks the ability to learn listwise relationships, and incorporating listwise data into its training would lead to degraded performance. For evaluation, we report MRR@10 on the MS MARCO development split (6,980 queries), and nDCG@10 on the TREC DL19 and DL20 passage ranking test sets, which contain 43 and 54 queries, respectively.

## 3.2 EXPERIMENTAL SETUP

All experiments are conducted on a machine equipped with four NVIDIA L20 GPUs (46 GB memory each). We choose **Qwen3-Reranker-4B** Yang et al. (2025) as our backbone model, which is a reranking base model trained on large-scale **text and code** corpora, for three reasons: (i) **scalability** – the 4B parameter size enables fast training/inference while retaining strong language understanding; (ii) **compatibility** – its autoregressive architecture aligns with our concatenation-based input format and supports efficient feature extraction for reranking; (iii) **stability** – Qwen models are well-validated in downstream fine-tuning, providing robust convergence for listwise objectives.

We conduct four main experiments: one overall comparison and three ablation studies.

1. **Passage Retrieval** We train LISTRANK on the same passage ranking training set as RankLLaMA and evaluate on the MS MARCO dev split as well as the DL19 and DL20 test sets. This experiment also compares against strong rerankers such as MONOBERT and CROSS-SIMLM.

2. **Attention Pool ablation** We replace the proposed attention pooling with the special-token hidden representation used in RankLLaMA to assess their relative effectiveness.

3. **MLP ablation** We compare our multi-layer perceptron (MLP) transformation module with the single fully connected layer adopted in RankLLaMA.

4. **ListRank Loss ablation** We replace the proposed ListRank loss with the standard InfoNCE loss to evaluate the contribution of the listwise objective. Because ListRank requires list-structured inputs to reveal its advantage, we augment the training data with MS MARCO passage-ranking lists for this experiment.

To mitigate GPU memory overflow, a common challenge in fine-tuning large language models, we apply LORA Hu et al. (2022) for efficient adaptation of the backbone while performing full-parameter tuning on the remaining network modules. Since listwise inputs include more documents than the pairwise format of RankLLaMA, we cap the maximum document list length at five to control memory usage. Each list contains two positive documents, two hard negatives, and one random negative, allowing the ListRank loss to learn fine-grained relevance ordering within each query-specific list.

## 3.3 PASSAGE RERANKING

The overall comparison experiment is conducted on the MS MARCO passage retrieval dataset. We evaluate **MRR** and **Recall@1k** on the development (dev) split, and **nDCG** on the DL19 and DL20 test sets. The comparative results are presented in Table 1.

On the dev split, the proposed ListRank model achieves MRR score of 45.0, surpassing RankLLaMA-7B by 0.1 while using only a 4B backbone—roughly 3/7 of the parameters. On the DL19 test set, ListRank obtains the best nDCG of 76.5, outperforming RankLLaMA-13B by 0.5. Compared with other LLM-based approaches such as RankGPT-4, which records an nDCG of 75.6 on DL19, ListRank also delivers stronger reranking performance. On the DL20 test set, ListRank again achieves the nDCG of 77.5, exceeding RankLLaMA-7B by 0.1, and likewise reaches state-of-the-art performance among all LLM-based models under the same compute scale. This contrast further highlights the superiority of the proposed ListRank algorithm in high-precision reranking.

## 4 ABLATION STUDIES

To validate the effectiveness of the proposed ListRank in improving retrieval performance, we conducted a dedicated ablation study. In this experiment, the **Base Model** adopts the same configuration as RankLLaMA. Concretely, RankLLaMA concatenates the query and document into a single sequence with task-specific separators and appends a closing token (e.g., "" in LLaMA-style tokenization). The hidden state of the final token serves as a single-vector summary of the input. A single fully connected (FC) layer maps this vector to a scalar relevance score. Training uses the InfoNCE objective over candidate lists to increase scores of relevant documents and decrease those of non-relevant ones. At inference, pointwise scores are computed for each $(Q, D_i)$ and candidates are sorted in descending order to produce the final reranking.

| Method | Model size | Source | top-k | DEV | | DL19 | DL20 |
|---|---|---|---|---|---|---|---|
| | | | | MRR@10 | R@1k | nDCG@10 | nDCG@10 |
| monoBERT (Nogueira et al. (2019)) | 110M | BM25 | 1000 | 37.2 | 85.3 | 72.3 | 72.2 |
| cross-SimLM (GLUE (2022)) | 110M | bi-SimLM | 200 | 43.7 | 98.7 | 74.6 | 72.7 |
| RankT5 (Zhuang et al. (2023)) | 220M | GTR | 1000 | 43.4 | 98.3 | – | – |
| RankLLAMA (Ma et al. (2023a)) | 7B | RepLLAMA | 200 | 44.9 | 99.4 | 75.6 | 77.4 |
| RankLLAMA-13B (Ma et al. (2023a)) | 13B | RepLLAMA | 200 | **45.2** | 99.4 | 76.0 | **77.9** |
| RankVicuna (Pradeep et al. (2023)) | 7B | BM25 | 100 | – | – | 66.8 | 65.5 |
| PRP (Qin et al. (2023)) | 20B | BM25 | 100 | – | – | 72.7 | 70.5 |
| RankGPT$_{3.5}$ (Sun et al. (2023)) | ? | BM25 | 100 | – | – | 65.8 | 72.9 |
| RankGPT$_4$ (Sun et al. (2023)) | ? | RankGPT$_{3.5}$ | 30 | – | – | 75.6 | 70.6 |
| **ListRank (Ours)** | 4B | RepLLAMA | 200 | 45.0 | **99.4** | **76.5** | 77.5 |

Table 1: The effectiveness of ListRank on the MS MARCO passage corpus compared to existing methods. Evaluation figures are copied from the original papers except for OpenAI Ada2 (from Lin et al., 2023).

### 4.1 ATTENTION POOL ABLATION

The attention pool ablation replaces the special-token hidden representation with the proposed attention pool module, which aggregates representations from all tokens while keeping all other settings identical to the base model. The model was trained on the MS MARCO passage ranking training set and evaluated on the dev, DL19, and DL20 test sets. The variant equipped with the attention pool is denoted as the attention base model. As shown in Table 2, on the DL19 test set, the base model reaches an nDCG of 66.81, whereas the attention pool base model increases it to 67.02. Similarly, on the DL20 test set, the nDCG improves from 64.99 to 65.85.

### 4.2 MLP ABLATION

To demonstrate that the proposed MLP-based feature transformation module outperforms a single-layer FC transformation, we conducted a second ablation experiment. Starting from the attention pool base model, we compared models using the MLP module versus a single-layer FC layer. The variant incorporating the MLP module is referred to as the MLP base model. Training and evaluation settings remained the same as above. As reported in Table 2, on the DL19 test set, the nDCG increases from 67.02 to 68.3, and on the DL20 test set, from 65.85 to 67.34.

### 4.3 LISTRANK LOSS ABLATION

To verify that the proposed ListRank loss provides stronger ranking supervision than InfoNCE, we performed a third ablation experiment. Based on the MLP base model, we compared training with ListRank loss against InfoNCE. The variant trained with ListRank loss is referred to as the ListRank base model. Training and evaluation followed the same MS MARCO passage ranking and dev/DL19/DL20 test settings. As summarized in Table 2, on the DL19 test set, the nDCG rises from 68.3 to 76.5 (+8.3), and on the DL20 test set, from 67.34 to 77.5 (+10.16).

The comparative experimental results show that the proposed ListRank model achieves superior text retrieval performance compared to LLM-based reranking approaches, while also attaining the highest retrieval metrics among pointwise reranking methods. Furthermore, the ablation studies demonstrate that each of the three proposed optimizations contributes positively to retrieval accuracy, providing additional evidence of the effectiveness of the proposed method. Among these optimizations, the ListRank loss delivers particularly notable performance gains, highlighting its strong capability to learn listwise relationships.

## 5 ANALYSIS

In addition to evaluating retrieval accuracy, we further investigate model convergence behavior and the effect of input sequence length on retrieval performance. Experimental results show that the proposed ListRank model exhibits faster and smoother convergence compared to all baselines,

| Model Variant | DL19 (nDCG@10) | DL20 (nDCG@10) |
|---|---|---|
| Base Model | 66.81 | 64.99 |
| + Attention Pool (Attention Base) | 67.02 | 65.85 |
| + MLP (MLP Base) | 68.3 | 67.34 |
| + ListRank Loss (ListRank Base) | **76.5** | **77.5** |

Table 2: Ablation results on the TREC DL19/DL20 test sets. nDCG is reported for both DL19 and DL20. All models share the same training data and backbone.

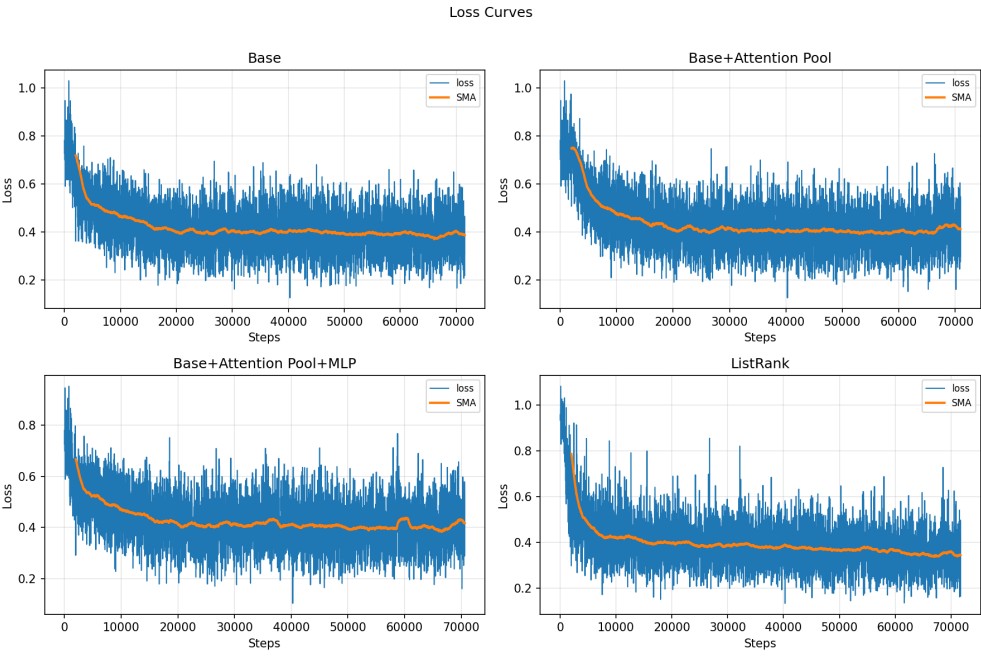

Figure 2: Training loss convergence curves for Base, Attention Base, MLP Base, and ListRank Base during fine-tuning.

indicating a more favorable learning curve and stronger ability to capture listwise relationships. Moreover, experiments on varying input length reveal that ListRank demonstrates clear advantages in long-text understanding, where longer input sequences lead to more significant performance gains by enabling more global semantic comprehension.

## 5.1 CONVERGENCE ANALYSIS

During both the overall and ablation experiments, we monitored training loss and plotted the loss curves to analyze convergence speed and stability. Figure 2 presents the loss trajectories of the Base, Attention Base, MLP Base, and ListRank Base models. The results reveal that each proposed optimization contributes to faster and smoother convergence. While the attention pool and MLP modules provide moderate improvements in convergence speed, the ListRank loss delivers a markedly stronger acceleration effect, resulting in both quicker and more stable convergence. Because the Base, MLP Base, and Attention Base models are trained with InfoNCE on pairwise data, they exhibit relatively small initial losses and a narrower gap between the initial and final losses. In contrast, the ListRank loss, which is trained on listwise data with longer list lengths, starts from a larger initial loss but converges to an even smaller final loss than the InfoNCE-based models, highlighting its strong capability to learn from listwise ranking relationships.

| Input Length | MRR@10 (dev) | nDCG@10 (DL19) | nDCG@10 (DL20) |
|---|---|---|---|
| 256 tokens | 44.5 | 76.21 | 76.61 |
| 512 tokens | 45.0 | 76.5 | 77.5 |

Table 3: Effect of input length on retrieval performance. Increasing the maximum input length from 256 to 512 tokens significantly improves both MRR and nDCG.

## 5.2 INPUT LENGTH ANALYSIS

In text retrieval and LLM-related tasks, the length of the input sequence often has a significant impact on performance. To examine this effect, we conducted experiments with varying input lengths. When increasing the maximum input length from 256 tokens to 512 tokens, the model achieved a clear performance boost, with MRR and nDCG each improving by approximately one point. These findings indicate that ListRank is well-suited for long-text scenarios, effectively leveraging the additional contextual information to yield substantial retrieval gains. The detailed results of the input-length experiments are summarized in Table 3.

## 6 CONCLUSION

In this work, we introduced ListRank, a listwise reranking framework designed for large-scale text retrieval. By integrating the attention pool module, the MLP module, and the ListRank loss, our approach strengthens representation learning while capturing global ranking relationships within candidate lists. Comprehensive experiments demonstrate that these components jointly accelerate convergence, stabilize optimization, and significantly improve retrieval effectiveness across standard benchmarks. Further investigations show that ListRank not only exhibits smoother and faster loss reduction but also benefits consistently from longer input sequences, enabling more comprehensive semantic understanding. Notably, the proposed ListRank loss equips LLM-based reranking models with listwise learning capabilities that far exceed those of existing methods, delivering particularly strong performance gains. These findings highlight the importance of explicitly modeling list-level dependencies and provide practical guidance for building efficient, high-performing retrieval systems.

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

# A  APPENDIX

## A.1  THE USE OF LARGE LANGUAGE MODELS (LLMS)

LLMs were used only for language polishing and not involved in research design, analysis, or results. The authors take full responsibility for the content.

