# OpenReview forum: "Fine-tuning large language models for text ranking with listwise constraints"
_ICLR.cc/2026/Conference — ICLR 2026 Conference Withdrawn Submission_

### Official Review · Reviewer_ytSp · 2025-10-29

**Soundness:** 2
**Presentation:** 2
**Contribution:** 2
**Rating:** 2
**Confidence:** 4

**Summary:**

This paper introduces a listwise fine-tuning method for LLM-based text reranking. The method improves three limitations of existing LLM rankers (single-token compression, shallow scoring heads, and pairwise objectives).

**Strengths:**

- Despite operating at 4B parameters, ListRank outperforms or matches 7B–13B rerankers (RankLLaMA, RankGPT), demonstrating accuracy-efficiency trade-off potential.
- The proposed list-wise training approach shows improvement based on the point-wise trained rerank models.

**Weaknesses:**

- The motivation for the architectural improvement is unclear. The authors need to further explain the limitations of single-token summarization. In addition, as I understand it, the introduced attention pooling and MLP essentially still produce a single-token summarization in the end. As shown in the ablation study, the performance gains brought by attention pooling and the MLP are also quite limited.
- The evaluation is conducted only on DL19 and DL20. The authors should evaluate on more benchmarks, such as BEIR.
- The authors need to provide more justification regarding the fairness and statistical significance of the comparisons. I noticed that the experimental setups for different models in Table 1 are not consistent.
- The novelty is limited. List-wise training has already been extensively studied in prior work. The baselines used in this paper are relatively weak; the authors should compare against more recent list-wise works [1,2,3].

[1] Reddy et al., FIRST: Faster Improved Listwise Reranking with Single Token Decoding, 2024, EMNLP

[2] Liu et al., Sliding Windows Are Not the End: Exploring Full Ranking with Long-Context Large Language Models, 2025, ACL

[3] Liu et al., Leveraging Passage Embeddings for Efficient Listwise Reranking with Large Language Models, 2025, WWW

- The writing quality needs improvement — many citations are directly glued to the text without proper formatting.

**Questions:**

- The explanation of the figure in the text is somewhat confusing. How exactly is a smoother convergence curve observed after adding attention pooling and the MLP?
- Why are point-wise and list-wise data mixed during training, and to what extent is the improvement simply due to using more training data compared to the baselines?
- Have the author compared against other list-wise methods such as ListNet?

---

### Official Review · Reviewer_5Vsk · 2025-10-31

**Soundness:** 2
**Presentation:** 2
**Contribution:** 2
**Rating:** 2
**Confidence:** 4

**Summary:**

This paper presents ListRank to address limitations in existing reranking approaches. The method includes three extra modules compared to the Qwen3-Reranker-4B backbone: (1) attention pooling, (2) a gated MLP, and (3) ListRank Loss. The model is trained on a RankGPT-refined subset of the MS MARCO passage ranking dataset. Experimental results show that ListRank achieves comparable performance on MS MARCO dev, TREC DL19, and DL20 benchmarks with a 4B model. Ablation studies confirm that each component contributes to performance.

**Strengths:**

1. The paper demonstrates each component's contribution with quantifiable improvements.

2. Comparable performance on standard benchmarks (MS MARCO, TREC DL19/DL20) with improved parameter efficiency.

3. Comprehensive analysis includes convergence behavior and input length effects.

**Weaknesses:**

This paper suffers from significant structural and organizational issues that require major revision. The novelty is limited, as both the gated MLP and listwise ranking losses are well-established methods in the literature. Additionally, the paper contains too many logical inconsistencies and typo errors throughout. Specific issues include:

1. Introduction Section Structure

The introduction reads more like a related work section, containing to many background without clearly describing the paper's motivation, solution, and contributions.

I suggest the authors restructure this section as follows:
Split the introduction into separate "Introduction" and "Related Work" sections
Start the introduction with the problem statement and key contributions
Move the historical survey (lines 24-51) to the Related Work section

2. Section 2.1 (Preliminaries) - Technical Writing Issues

Line 75: "assigns each candidate D_i a real-valued relevance score, s_i, and sorts the candidates..." → The notation s_i should be introduced here.

Line 76: "Collecting scores over the candidate list yields s = [s1, s2, . . . , sk], which induces a permutation by sorting in descending order" lacks a proper verb structure. Suggested revision: "Collecting scores over the candidate list yields s = [s1, s2, . . . , sk], which induces a permutation when sorted in descending order."

Line 86: "(pointwise formulation)" and "(listwise formulation)" should reference actual mathematical formulations rather than being mere textual labels. Consider providing brief formula definitions or removing these parentheticals if they are not formally defined.

Lines 87-88: This content is redundant with lines 80-84 and should be removed or consolidated.

3. Section 2.2 (Base Model) - Citation Format

Line 107: "query–document pair (Section 2.1)" should use \ref{sec:preliminaries} instead of manual section numbering for proper LaTeX cross-referencing.

4. Section 2.4 (Multi-Layer Perceptron) - Content Focus Issues
Overly verbose general MLP description: The MLP description is too generic and well-known. Given that Transformers already contain MLP layers in each block, this extended justification is unnecessary and should be significantly condensed.

Misaligned emphasis: This section should focus on the gated structure innovation rather than rehashing basic MLP concepts. The gated mechanism (up-projection + gate-projection with element-wise multiplication) is the key contribution and deserves more emphasis.
Notation inconsistency: The formula for h_out should use superscript notation (h^{out}) to maintain consistency with Section 2.5.

5. Section 2.5 (ListRank Loss) - Structural Issues

Lines 249-250 misplaced: The sentence "Based on the Attention Pool, MLP optimization, and ListRank loss proposed in this work, we construct a new model, which we refer to as ListRank" should appear to the beginning of Section 2 or immediately after Section 2.1 as an overview, not at the end of the loss description.

Lines 251-252 redundancy: These lines repeat information already conveyed and should be removed or merged with the previous paragraph.

Sections 3 and 4 also contain typographical errors and require careful revision.

**Questions:**

1. In Figure 2 (Training loss convergence curves), I noticed that Base+Attention Pool exhibits a loss spike between steps 65,000-70,000, and Base+Attention Pool+MLP shows another spike between steps 60,000-70,000. What do you think causes these spikes?

2. Regarding Section 3.2 (EXPERIMENTAL SETUP), could you clarify how the random negatives are sampled during data construction?

3. The results show that adding ListRank Loss on top of Attention Pool + Gated MLP yields approximately 8% improvement. What performance would we see if we trained Base + ListRank Loss directly, without the attention pooling and MLP modules?

---

### Official Review · Reviewer_t7gj · 2025-10-31

**Soundness:** 2
**Presentation:** 3
**Contribution:** 2
**Rating:** 4
**Confidence:** 3

**Summary:**

This paper proposes ListRank, a new framework designed for large language model (LLM)-based text retrieval and reranking tasks. The main contribution lies in addressing limitations of current LLM-based reranking approaches through three key innovations: A customized attention-based fusion of token-level representations. A multi-layer perceptron (MLP) module for enhanced feature transformation. A ListRank loss designed to model listwise ordering, thereby improving the fine-grained relevance order of candidate documents in a ranking task. The experimental results on MS MARCO and TREC datasets show that ListRank outperforms existing state-of-the-art reranking models in terms of mean reciprocal rank (MRR) and normalized discounted cumulative gain (nDCG) at 10.

**Strengths:**

1. The combination of attention pooling, MLP for feature transformation, and a listwise loss function (ListRank) represents a novel approach in the context of LLM-based text reranking.

2. The paper presents a detailed and well-structured approach to a complex problem in retrieval. The proposed ListRank loss is a creative contribution, addressing the underutilization of global ordering in most existing reranking methods.

3. The presentation of the method is clear, with well-defined steps outlined in the methodology section. The ablation study provides a strong demonstration of the contribution of each component, which adds to the clarity of the paper’s argument.

**Weaknesses:**

1. The paper presents the ListRank loss in a practical context but lacks a deep theoretical justification for why the method should be superior to existing pairwise or pointwise methods. More rigorous analysis of the listwise nature of the problem and how exactly ListRank capitalizes on this could strengthen the argument.

2. The experiments, while thorough, seem primarily focused on empirical validation without enough exploration into the why behind the observed performance improvements. For instance, why does ListRank perform better in long-text retrieval, and what exactly about the MLP and attention fusion modules leads to better results? A deeper analysis would have added significant value.

3.  The use of MS MARCO as a training dataset is well-established but also well-trodden. The paper could have explored other more challenging or domain-specific datasets to better illustrate the broader applicability of the method. Furthermore, it's unclear if the ListRank approach could generalize to domains outside of traditional information retrieval.

**Questions:**

1. The paper mentions that ListRank outperforms existing models in terms of retrieval metrics like nDCG. How does ListRank handle cases where candidate documents in the top-k list are of significantly different lengths? Could performance degrade with extremely long documents or highly diverse types of content?

2. The authors claim that ListRank’s attention pooling module resolves the hallucination problem in long-text scenarios. Can the authors provide a more detailed explanation of how ListRank mitigates hallucinations better than existing models? Is there a theoretical framework behind this, or is it purely experimental?

3. Could the authors elaborate more on the scalability of ListRank when applied to extremely large datasets or when fine-tuning large language models on a distributed setup? What specific challenges arise in such cases, and how does the model handle them?

---

### Note · Authors · 2025-11-17

I have read and agree with the venue's withdrawal policy on behalf of myself and my co-authors.